# A New Fluorescent Probe for Hydrogen Sulfide Detection in Solution and Living Cells

**DOI:** 10.3390/molecules28176195

**Published:** 2023-08-23

**Authors:** Wei Feng, Qicai Xiao, Lu Wang, Yuanyong Yang

**Affiliations:** 1School of Pharmacy, Guizhou Medical University, Guiyang 550025, China; 2BGI-Shenzhen, Shenzhen 518038, China; 3School of Pharmaceutical Sciences (Shenzhen), Sun Yat-sen University, Shenzhen 510006, China; xiaoqicai@mail.sysu.edu.cn; 4Department of Physiology, College of Basic Medicine, Guizhou Medical University, Guiyang 550025, China; wanglu@gmc.edu.cn

**Keywords:** hydrogen sulfide, fluorescent probe, H_2_S selective detection, H_2_S quantification, azide-reduction

## Abstract

Since Hydrogen Sulfide (H_2_S) was recognized as a gas transmitter, its detection and quantification have become a hot research topic among chemists and biologists. In this area, fluorescent probes have shown great advantages: fast and strong response, low detection limit and easy manipulation. Here we developed a new fluorescent probe that detected H_2_S selectively among various bioactive and inorganic salts. This probe was based on the core structure of fluorescein and reacted with H_2_S through azide-reduction. Great linearity was achieved correlating fluorescence intensity and H_2_S concentrations in solution. The detection of H_2_S in cancer cells was also achieved.

## 1. Introduction

Since being recognized as an important cell-signaling gasotransmitter [1,2], hydrogen sulfide (H_2_S) has attracted great interest from research areas of both biology [3,4] and chemistry [5,6]. H_2_S has been reported playing important roles in many physiological and pathological conditions, such as inflammation regulation [7,8], cardio protection [9], neuromodulation [1], hypertension [10], pain perception [11,12] and even cancer [13]. It has been found that the amount of endogenously generated H_2_S varies in different tissues. The H_2_S amount in healthy cells is also different from that in abnormal cells. As a result, detection and quantification of H_2_S are of great importance in understanding its biological effects. Various technologies and methods have been developed for H_2_S detection [14,15]. Among these technologies, fluorescent probes show great potential both in vitro and in vivo, due to fast response, excellent selectivity, high sensitivity and real time imaging [16,17,18]. Several strategies have been developed for the design of fluorescent probes of H_2_S [19], including the reduction of aryl azide [20] and aryl nitro compounds [21,22], H_2_S-specific reactions based on its nucleophilicity [23], an H_2_S-induced metal displacement approach [24], and a disulfide exchange reaction [25]. Among these strategies, azide-reduction has attracted extensive attention due to its fast reaction rate, high selectivity and harmless byproduct N_2_. In 2011, the Chang group reported a series of rhodamine-based fluorescent probes (SF1 and SF2), which successfully and selectively detect H_2_S via azide reduction in both water and living cells [26]. In the same year, the Wang group developed a fluorescent probe DNS–Az, which reached the maximum fluorescent intensity within 30 s upon mixing with sulfide anion [27]. The extremely fast reaction was believed to favor the real time detection of H_2_S in vivo. At the same time, various fluorophores were modified with the azide group for the design of H_2_S probes, such as Cyanines–N_3_ (Cy–N_3_) [28], Phenanthroimidazole–N_3_ (PI–N_3_) [29], 7-nitrobenz-2-oxa-[1,3]diazole–N_3_ (NBD–N_3_) [30], Pyrene-1,3,6-trisulfonate–N_3_ [31], Coumarin–N_3_ [32,33], Naphthalimide–N_3_ [34] and so on. In all these probes, an azide group was attached to an aromatic ring or a conjugate system.

Recently, the mechanism for a sulfide-induced azide reduction was studied experimentally and computationally in detail by Henthorn [35]. The active species HS^−^, which is the dominant form of H_2_S in aqueous solution, firstly attacks the tailed N atom of the azide group. Then another molecule of HS^−^ comes in to attack the sulfur atom, forming HS_2_^−^ and the release of one molecule of N_2_ (Appendix A).

Azidomethyl has been used to protect amine or alcohol in the synthesis of carbohydrates [36,37], nucleosides [38], triazoles [39] and peptides [37]. Nowadays, the most important application of azidomethyl is as a reversible terminator in DNA sequencing by synthesis (SBS) [40]. The azidomethyl capped 3′–OH can be easily and quickly deprotected with reducing agents, and thus, the SBS process can be continued. Because DNA sequencing is such a big project and it takes a long time, the chemical reaction of each step is required to be as fast as possible. Azidomethyl reduction meets all requirements perfectly. Inspired by these works, we developed a turn-on fluorescent probe (FL–N_3_) for H_2_S detection by protecting fluorescein with azidomethyl, and tested the probe in aqueous solution and cancer cells.

## 2. Results

### 2.1. Synthesis

The probe FL–N_3_ was based on fluorescein with one of the hydroxyl groups protected by a polyethylene glycol (PEG) side chain, which was expected to increase the probe’s hydrophilicity [41]. The other hydroxyl group was caged with azidomethyl, which reacts with sulfide to release free hydroxyl and turn fluorescence on (Figure 1, FL–N_3_ to FL–ONa). We synthesized this probe starting from commercially available fluorescein sodium (compound A) in five steps including two processes of two–step in one–pot. Fluorescein sodium and tosylated hexaethylene glycol monomethyl ether were heated together in DMF. The free carboxylic acid and hydroxyl were both protected with hexaethylene glycol monomethyl ether by forming an ester and an ether, respectively. Without purification of the intermediate, hydrolyzation of the newly formed ester with sodium hydroxide and subsequent acidification with HCl led to the formation of compound B. The phenolic hydroxyl was protected with chloromethyl methyl thioether (as in compound C), which was then converted to chloromethyl by reacting with NCS and TMSCl. The probe (FL–N_3_) was achieved after nucleophilic substitution of the chloride with azide. Although five steps were required, two intermediates were not necessarily to be separated, which made the synthesis quite straight forward.

### 2.2. Fluorescence Response

We first tested the UV absorbance of the probe in solution with and without Na_2_S. The probe alone showed no absorbance in the range from 300 to 600 nm. After being incubated with sodium sulfide for 30 min, a strong absorbance peak appeared at 455 nm (Appendix A), which indicated the formation of fluorescent product. The fluorescence properties of this probe were then tested at an excitation wavelength of 455 nm. Although a PEG side chain has been attached, the solubility of the probe is still not good enough in pure water. As a result, tests were carried out in water and a methanol mixture. When the probe FL–N_3_ was mixed with Na_2_S at a concentration of 1 mM, the fluorescence intensity increased by 32-fold at λ_em_ of 515 nM after 20 min (Figure 1a). The fluorescence response over time was also examined (Figure 1b). FL–N_3_ and Na_2_S were mixed at concentrations of 5 mM and 10 mM, respectively. The fluorescence intensity increased quickly in the first twenty minutes. It reached the max after a slow increase during the period from 20 to 60 min. The maximum fluorescence intensity kept almost consistent for one hour from 60 to 120 min. The long-lasting fluorescence of the probe could be important in many applications [42]. HPLC analysis demonstrated the formation of a major product (Appendix A) and MS analysis indicated that the fluorescence increase was due to the formation of FL–ONa (Figure 1, mass spectra data in Appendix A).

### 2.3. Selectivity

To test the probe’s selectivity, we selected a series of biothiols and bio-relevant sulfur-containing anions. Only HS^−^ induced a significant increase of fluorescence (Figure 2). Biothiols including cysteine, glutathione and dithiothreitol led to negligible fluorescence intensity change. Bio-relevant reducing salts such as sodium L–ascorbate, sodium thiocyanate and sodium thiosulfate also failed to turn the probe on. Other inorganic salts that were commonly used in buffer were also tested (Appendix A). Ammonium acetate, sodium citrate, lithium chloride and dipotassium phosphate did not increase the fluorescence intensity. However, basic salts like potassium carbonate (pH 10.3) and lithium hydroxide (pH 10.7) led to about three- and five-fold fluorescence increases, respectively, compared with the blank. This is probably due to the acidic property of the C–H bond on the carbon of the azidomethyl group, which was deprotonated by hydroxide. One molecule of nitrogen and formamide were released to liberate the free hydroxyl group. Under basic conditions (pH 10.3~10.7), the lacton was transformed into monoaion to turn on the fluorescence. A possible mechanism was predicted, as in Appendix A.

### 2.4. Linearity

It has been found that the bioactivity of H_2_S is concentration-dependent [43], which means the quantification of H_2_S in solution or cells is of great importance in H_2_S study. When FL-N_3_ was exposed to Na_2_S of different concentrations, a linear correlation was observed between fluorescence responses of the probe and Na_2_S concentrations in the range of 0 to 0.8 mM, enabling a calibration equation to be established and applicable to the analysis of H_2_S concentrations (Figure 3). The linear equation was found to be Y = 4584.3X + 204.66. R^2^ = 0.9952. The detection limit was calculated to be 3.99 μM based on the signal-to-noise ratio (S/N = 3).

### 2.5. Detection in Cells

Since the probe FL–N_3_ has well-behaved fluorescent properties and selectivity for H_2_S, it is probably suitable for the detection of H_2_S in cultured cells. Confocal imaging to visualize H_2_S inside cells was carried out. MCF–7 cells were incubated together with FL–N_3_; after washing free molecules of the probe away, Na_2_S solution was added. Confocal images were taken after another 60-min incubation and results are shown in Figure 4. The probe successfully entered cells and reacted with sulfide to show red fluorescence (Figure 4a). In contrast, the sample of control (Figure 4b) showed no fluorescence at all. We also tested compound D, which contains a methyl group instead of a PEG side chain. Surprisingly, compound D failed to enter cells and little fluorescence was observed after treatment with Na_2_S (Figure 4c). This result proved the importance of the PEG side chain for the probe’s hydrophilicity and cell permeability, which are key factors in bio-applications.

## 3. Discussion

We designed and synthesized a turning-on probe for the detection of H_2_S in solution and in cells. The probe works through the reduction of the azidomethyl group by H_2_S. It also inspired us that the azidomethyl group could be an ideal choice in the modification of turning-on dyes. When fluorescence is turned off by capping different dyes with azidomethyl, it can be turned on easily by adding certain reductant, which could have potential in biology research.

## 4. Materials and Methods

Procedure for probe response to sodium sulfide test: The probe FL-N_3_ was dissolved in methanol at a concentration of 2 mM. Sodium sulfide was dissolved in water at a concentration of 40 mM. Two wells of a corning 96-well plate were selected and to each well was added 100 μL probe solution. Then, 100 μL Na_2_S solution was added to one of the wells and to the other was added 100 μL water. The plate was incubated at RT for 20 min before reading fluorescence on a plate reader. The fluorescence was read at excitation wavelength λ_ex_ = 455 nm with emission wavelength λ_em_ set from 500 nm to 600 nm.

Procedure for fluorescence response over time test: The probe FL-N_3_ was dissolved in methanol at a concentration of 10 mM. Sodium sulfide was dissolved in water at a concentration of 20 mM. To a corning 96-well plate was added 100 μL probe solution and 100 μL Na_2_S solution. The fluorescence was read on a plate reader every minute in 2 h with excitation wavelength λ_ex_ = 455 nm and emission wavelength λ_em_ = 515 nm.

Procedure for probe selectivity test: The probe FL-N3 was dissolved in methanol at a concentration of 1 mM. Other tested compounds or salts were dissolved in water at a concentration of 10 mM. To a corning 96-well plate was added 160 μL water and 20 μL probe solution. Then 20 μL aqueous solution of one tested compound or salt was added to one well. To the well of the blank, 20 μL water was added. After 30-min incubation at RT, fluorescence was read on a plate reader (brand) with excitation wavelength λ_ex_ = 455 nm and emission wavelength λ_em_ = 515 nm.

Procedure for linear correlation test: The probe FL-N_3_ was dissolved in methanol at a concentration of 10 mM. To 6 wells of a corning 96-well plate was added 20 μL of the probe solution. Following this, to the 6 wells was added 180 μL, 160 μL, 140 μL, 100 μL, 60 μL, 20 μL water in order. Then 0 μL, 20 μL, 40 μL, 80 μL, 120 μL, 160 μL of sodium sulfide solution (1 mM) was added to the 6 wells in order. After incubation at RT for 20 min, the fluorescence was read on a plate reader with excitation wavelength λ_ex_ = 455 nm and emission wavelength λ_em_ = 515 nm.

Procedure for confocal imaging: Human breast cancer (MCF7) cells were purchased from Biobw Pte Ltd. (ATCC, HTB22). MCF7 cancer cells were grown in RPMI Media (Merck, CN), which was supplemented with 10% fetal bovine serum, 50 μg/mL penicillin and 50 μg/mL streptomycin at 37 °C, 5% CO_2_. The cells were subcultured to 80–90% confluency and used within 15–20 passages for the assay. MCF7 cells were seeded at 20,000 cells/well in an 8-chamber plate (0.8 cm^2^). Probe FL-N_3_ (20 μM in DMSO) was added and cells were incubated for 2 h at 37 °C. Cells were then washed with PBS (3×). Then Na_2_S solution (0.1 mM) was added and incubated for 5 h. Fluorescent images of the live cells were obtained using a confocal fluorescence microscope FV 1000 and processed using Olympus Fluoview Ver.3.1. Viewer (Olympus, Tokyo, Japan).

Procedure for the synthesis of compound B: Fluorescein sodium salt (135 mg, 0.36 mmol) was added to a two necked rbf; after evacuation, the rbf was recharged with N_2_, and anhydrous DMF (3 mL) was added, followed by the addition of 2,5,8,11,14,17-hexaoxanonadecan-19-yl 4-methylbenzenesulfonate (450 mg, 1 mmol). The mixture was heated to 90 degrees centigrade and stirred for 36 h. The mixture was then cooled to room temperature, and DMF was removed by evaporation and the residual was re-dissolved in 10 mL 10% NaHCO_3_ solution. The residual was extracted with EA for three times and the organic layer was combined, washed with brine, dried over Na_2_SO_4_ and concentrated. The residual was re-dissolved in 5 mL methanol, 2 M NaOH (2 mL) was added, and stirred at room temperature for 2 h. Then 1 M HCl was added slowly to adjust pH to 2, and the mixture was stirred for 30 min. Methanol was removed via rotary evaporation and the residual was extracted with EA for three times. The combined organic layer was washed with brine, dried over Na_2_SO_4_ and concentrated. The product was isolated via silica gel chromatography (DCM/MeOH = 100/1) with a yield of 65%.

Procedure for the synthesis of compound C: Compound B (122 mg, 0.2 mmol) was dissolved in 10 mL anhydrous MeCN under Ar. Silver oxide (92 mg, 0.4 mmol) was added, followed by the addition of chloromethyl methyl thioether (167 μL, 2 mmol) and one drop of pyridine. The mixture was heated to 50 degrees centigrade and stirred for 48 h. The mixture was cooled to room temperature and solvent was removed via rotary evaporation. The product was isolated via silica gel chromatography (DCM/MeOH = 100/1) with a yield of 81%.

Procedure for the synthesis of probe FL-N_3_: Compound C (67 mg, 0.1 mmol) was dissolved in 5 mL anhydrous DCM under Ar. N-chlorosuccinimide (16 mg, 0.12 mmol) was added and the mixture was stirred at room temperature for 3 h. Then trimethyl silane chloride (13 mg, 0.12 mmol) was added and the mixture was stirred at room temperature further for 6 h. After removal of solvent via rotary evaporation, the residual was dried under vacuum and re-dissolved in dry THF under Ar; following this, TMSN_3_ (0.2 mmol) and TBAF (0.2 mmol, 1 M in THF) were added and the mixture was stirred at room temperature overnight. The solvent was removed via rotary evaporation and the product was isolated via silica gel chromatography (DCM/MeOH = 100/1) as a yellow oil. The product was further purified via prep HPLC in case it was impure.

## 5. Conclusions

In summary, we have developed a new strategy to construct a fluorescein-based probe for the detection of H_2_S in solution and in cells. The probe showed fast response and excellent selectivity to sulfide over a series of bio-relevant sulfur-containing compounds and inorganic salts. The linear correlation of Na_2_S concentration and probe fluorescent response made the quantification of H_2_S in solution possible. Its successful application in cell confocal imaging may provide a new tool in the study of the biological effects of hydrogen sulfide.

## Data Availability

The data presented in this study are available in the Appendix A.

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
