# Peer review of "A New Fluorescent Probe for Hydrogen Sulfide Detection in Solution and Living Cells"

_molecules, 2023, doi:10.3390/molecules28176195_

Round 1
Reviewer 1 Report
Feng et al. report a new fluorescent probe based on fluorescein skeleton for the detection of hydrogen sulfide. The aim of the research is clear but some things need to be improved.
There are major shortcomings, which needs to be addressed by the authors, as listed below:
1. One of the crucial parameters controlling the performance of any probe used for the detection of hydrogen sulfide is the rate constant of the reaction between the probe and analyte. The authors should provide the kinetic data reported for the direct analyses or obtained by competition kinetics.
2. The reaction of the probe with hydrogen sulfide should be studied by following the substrate consumption and the formation of corresponding products using the HPLC technique. Identification and quantitative analyses of the products of the reaction of the probe with hydrogen sulfide should be based on comparison with synthesized authentic standards and/or mass spectrometric analyses (as shown before – Dyes and Pigments 196 (2021) 109765).
3. The molar absorption coefficients and fluorescence quantum yields of the synthetized compounds (FL-N3 and Fl-ONa) should be added in the manuscript. These data, along with the location of the absorption and emission bands, can be summarized in a table.
4. The article should include how the MCF-7 cells were cultured.
5. It is worth to test the cytotoxicity of the FL-N3 and Fl-ONa (e.g. using the MTT test).
6
6. The language of the manuscript requires extensive copy-editing - it is difficult to follow the authors' logic due to the grammar errors. There are also many editing errors that need to be corrected. E.g. line 12 – “responds”; line 21 – “The introduction”; line 70 – “hydrolysis”; Materials and Methods – “λex”, “λem”; “degree”; “PH”.
Reviewer 2 Report
The manuscript ‘A New Fluorescent Probe for Hydrogen Sulfide Detection in Solution and Living Cells’ by Feng et al. described a new fluorescent probe for the detection of hydrogen sulfide both in solution and in living cells. Also, the competitive responses of different biologically relevant sulfides and inorganic sulfides (H2S) towards the developed probe are being evaluated. The probe is showing a fast response and excellent selectivity to inorganic sulfide (Na2S) only. Therefore, the probe may be considered a valuable tool for studying the consequences of hydrogen sulfide in a biological specimen. Overall, it seems to be an important finding which may have potential applications in further bio-analytical research and drug discovery.
However, before publication, the manuscript needs to be revised to improve the quality of the work done and the appearance of the manuscript as well.
1. Figures 1 a and 1 b are quite blurred in appearance and also fonts are too small to read. Authors should improve the quality of these figures.
2. The mechanistic aspect of the reaction between the probe and Na2S is not described anywhere in the manuscript. It should be taken care of by the author.
3. Although the course of change in fluorescence intensity with time has been given the corresponding spectra should be included in the manuscript. Also, Fluorescence responses should be checked in different solvents and in different water ratios.
4. Since the probe is being employed for the detection of H2S in cultured cells, the cytotoxicity of the probe should also be checked, and also the amount of H2S present in the cell should be quantified and justified by comparing the data with commercial H2S probe/maybe with the literature precedence.
5. Since the amount of H2S varies in different cell lines, authors should explore the method in different cell lines too.
6. Authors did not talk about the sensitivity of their probe. At least the lower limits of their probe in detecting H2S should be explored.
Reviewer 3 Report
The manuscript in question seems to me to have been prepared without much care. This is the opinion I get just by reading the first line of the introduction, a sentence that makes no sense. Later, when I tried to download the Supporting Information, the website provided by the ms. did not exist, although at my request, the assistant editor contacted the authors and was able to provide me the requested document. Finally, the references are not updated. Indeed, only reference 6 is a recent review on NIR probes for H2S sensing. All other references dealing with H2S sensing are from 2015 or earlier, while the development of H2S-sensitive fluorescence probes is one of the most rapid growing areas in the field of H2S biology. More recent reviews on H2S are: H. Ibrahim et al. Journal of Advanced Research 27 (2021) 137–153, Sun Young Park, Shin A Yoon, Y. Cha et al. Coordination Chemistry Reviews 428 (2021) 213613 and H. Li, Y. Fang, J. Yan et al. Trends in Analytical Chemistry 134 (2021) 116117, that might be of interest to the authors. A novel xanthene-based colorimetric and fluorescence probe for detection of H2S in living cells (J. Liu et al. J. Luminesc. 2018, 204, 480) may also be one appropriated reference in the ms.
In general, a good fluorescent probe should react rapidly under physiological conditions without the need for organic solvents to monitor changes in H2S fluxes in living cells in real time. Moreover should emit in the near-infrared to achieve greater tissue penetration, causes less cellular photo-damage and minimizes the interference from auto-fluorescence. Xanthene dyes possess outstanding spectroscopic and photophysical properties, but they also have some limitations as the low sensitivity and long-time exposure needed to catch the changes of H2S levels. Additionally and crucially, autofluorescence of tissues can disturb the use of probes having a similar excitation/emission wavelength. As the proposed new probe needs a relatively long time to reach maximum fluorescence intensity and the probe emission coincides with the autofluorescence of biological fluids, cells and tissues, the authors should cite the highlights of the proposed probe that justify their publication in the Journal MOLECULES.
In addition, the probe should show high selectivity and not to interact with other endogenous antioxidants. Therefore, although the authors show the specificity of the probe in Figure 2, the study should be extended to other antioxidants such as NADH, NADPH.
As indicated in Scheme 1, the cause of fluorescence is the monoanion of fluorescein, which is consistent with Figures 1a and S1. This should be so because the lactone neutral form seems to exist only in the presence of organic solvents (Lindquist, L. Ark. Kemi, 1960, 16, 79) and would not be the majority form only in a 10% methanol solvent. In Yguerabide et al, Photochem. Photobiol. 1994, 60, 435, a study on the prototropic forms of fluorescein in solution and the pK of the different equilibria is presented. For the neutral-monoanion equilibrium of fluorescein in aqueous solution the pKN is 4.4 according to Lindquist, or 4.24 according to Diehl (Diehl, H. Talanta, 1989, 36, 413). Since the authors do not provide data on the working pH and due to the CO2 dissolved in the water, it can be assumed that, in the medium used, there is an equilibrium between the neutral form and the monoanion. For this reason, when basic salts are added (lines 110 and 111), the proportion of monoanion, which has a higher molar extinction coefficient and quantum yield, increases, thus obtaining greater fluorescence. Therefore it seems to me that the mechanism is simpler than the one proposed in Figure S5. Since if the dianion of fluorescein was produced, the increase in fluorescence would be much greater than that observed. Due to the previous discussion, the authors must indicate the pH of the medium in which they carried out their experiments.
Round 2
Reviewer 1 Report
Accept in present form
Author Response
We thank reviewer very much for accepting this manuscript for publication in molecules.
Author Response
We thank reviewer very much for accepting this manuscript for publishing in molecules.
Reviewer 3 Report
The authors explain that their probe is highly selectivity towards H2S among a lot of salts and bio-antioxidants. However, when I have suggested that they extend their study on specificity to other antioxidants such as NADH, NADPH... the answer is that they need to synthesize the probe again and that they will add the data obtained as soon as possible.
I understand that until these data are included in a new version, the manuscript should not be published.
In their response, the authors assert that application of the probe for H2S quantification and detection in cells also showed the probe’s potential. However, they have not carried out any study on the toxicity of the probe, which when dissolved in water/methanol, can be too toxic to use.
In lines 113 and 114 the authors establish that basic salts like potassium carbonate and lithium hydroxide led to about three– and five–fold fluorescence increase respectively comparing with the blank. In my first review I tried to help them to a possible interpretation of these results based on the pH of the solutions and the pKa of fluorescein. Although in their answer they seem to accept that a greater amount of monoanion originates in such basic solutions, that discussion has not been carried over to the new version of the ms. By the way, pH is never written as PH.
